# Imipramine Inhibits Migration and Invasion in Metastatic Castration-Resistant Prostate Cancer PC-3 Cells via AKT-Mediated NF-κB Signaling Pathway

**DOI:** 10.3390/molecules25204619

**Published:** 2020-10-11

**Authors:** Eun Yeong Lim, Joon Park, Yun Tai Kim, Min Jung Kim

**Affiliations:** 1Research Group of Functional Food Materials, Korea Food Research Institute, 245, Nongsaengmyeong-ro, Iseo-myeon, Wanju-gun, Jeollabuk-do 55365, Korea; 50005@kfri.re.kr (E.Y.L.); 50029@kfri.re.kr (J.P.); 2Department of Food Biotechnology, Korea University of Science & Technology, 217 Gajeong-ro, Yuseong-gu, Daejeon 34113, Korea; 3Research Group of Natural Materials and Metabolism, Korea Food Research Institute, 245, Nongsaengmyeong-ro, Iseo-myeon, Wanju-gun, Jeollabuk-do 55365, Korea

**Keywords:** imipramine, antidepressants, prostate cancer, migration, invasion

## Abstract

Imipramine (IMI) is a tricyclic synthetic antidepressant that is used to treat chronic psychiatric disorders, including depression and neuropathic pain. IMI also has inhibitory effects against various cancer types, including prostate cancer; however, the mechanism of its anticancer activity is not well understood. In the present study, we investigated the antimetastatic and anti-invasive effects of IMI in metastatic castration-resistant prostate cancer PC-3 cells, with an emphasis on the serine/threonine protein kinase AKT-mediated nuclear factor kappa B (NF-κB) signaling pathway. While IMI did not induce cell death, it attenuated PC-3 cell proliferation. According to the wound healing assay and invasion assay, migration and invasion in PC-3 cells were significantly inhibited by IMI in a dose-dependent manner. IMI significantly downregulated p-AKT protein expression but upregulated phospho-extracellular signal-regulated kinase (ERK1)/2 protein expression levels. Furthermore, IMI treatment resulted in decreased AKT-mediated downstream signaling, including p-inhibitor of κB kinase (IKK)α/β, p-inhibitor of κB (IκBα), and p-p65. Inhibited NF-κB signaling reduced the secretion of several proinflammatory cytokines and chemokine by PC-3 cells. Overall, our study explored the negative correlation between the use of antidepressants and prostate cancer progression, showing that IMI attenuated cell viability, migration, and invasion of PC-3 cells by suppressing the expression of AKT and NF-κB-related signaling proteins and secretion of tumor necrosis factor-*α* (*TNF-α*), interleukin-1β (*IL-1β*), and monocyte chemoattractant protein-1 (*MCP-1*).

## 1. Introduction

Prostate cancer is one of the most common invasive cancers in men, with 1,276,000 cases and 359,000 deaths worldwide in 2018 [1]. In Korea, the incidence of prostate cancer has more than doubled in the last 15 years due to the aging population and the westernization of eating habits [2]. Patients with localized prostate cancer are managed by radiation therapy, surgery, or hormone therapy, while locally advanced prostate cancer and metastatic prostate cancer are mainly treated with androgen-deprivation therapy (ADT) [3]. ADT leads to the remission of prostate cancer in about 90% of the patients as evidenced by decreased circulating levels of prostate-specific antigen (PSA) [4]. However, continuous ADT treatment attenuates the response of metastatic castration-sensitive prostate cancer (mCSPC) to ADT within 12–24 months and causes changes in the types of metastatic prostate cancer from mCSPC to metastatic castration-resistant prostate cancer (mCRPC) [5,6]. mCRPC is associated with a high risk of morbidity and mortality, with a short survival time of only 16–18 months [7]. Therefore, it is urgent to suppress mCRPC and develop new efficient chemotherapeutic agents for the treatment of ADT-resistant mCRPC.

Food and Drug Administration (FDA)-approved chemotherapeutic drugs for mCRPC are mitoxantrone, docetaxel, and cabazitaxel. Mitoxantrone, approved in 1996, had a palliative benefit and decreased PSA levels [8,9,10]. Subsequently, docetaxel and cabazitaxel, approved in 2004 and 2010, respectively, replaced mitoxantrone; these drugs relieved symptoms, improved PSA response rate, and led to extended overall survival in mCRPC patients [9,11,12,13]. However, both of these drugs also have limitations because some patients with mCRPC are not controlled by these drugs, and resistance to treatment eventually occurs. To overcome these disadvantages, researchers are constantly exploring new chemotherapeutic agents.

One of the suggestions that resulted from these efforts is to use antidepressants as new mCRPC inhibitors. Several classes of antidepressants exist, such as nonselective monoamine reuptake inhibitors, selective serotonin reuptake inhibitors, nonselective monoamine oxidase inhibitors, monoamine oxidase A (MAO-A) inhibitors, and other antidepressants including tricyclic antidepressants (TCAs) [14,15]. Among these, MAO-A inhibitors are known for their efficacy and mechanisms of prostate cancer suppression [16,17]. MAO-A inhibitors, including clogyline and mocrobemide, suppressed the activation of Shh-interleukin 6-receptor activator of nuclear factor kappaB ligand (Shh–IL6–RANKL) signaling, further suppressed the metastasis of CRPC cells, and prolonged overall survival in patients in preclinical studies [16]. In addition, antiandrogen enzalutamide-resistant mCRPC cells derived by chronic exposure to enzalutamide could be re-sensitized to enzalutamide by treatment with MAO-A inhibitors such as phenelzine and clorgyline [17]. However, the efficacy and mechanisms of action of TCAs in mCRPC cells have been poorly studied.

Imipramine (IMI, 10,11-dihydro-*N*,*N*-dimethyl-5*H*-dibenz[*b,f*]azepine-5-propanamine hydrochloride) is a member of the TCA family and is used in the treatment of depressive disorders, neuropathic pain, and nocturnal enuresis [18,19,20]. In addition to these neurologic properties, IMI has exhibited antiproliferative and anticancer activities in various cancer cells such as myeloma, small-cell lung cancer, pancreatic neuroendocrine cancer, and prostate cancer cells [21,22,23]. However, the mechanism of IMI action in prostate cancer, especially mCRPC, has not been fully studied. The only known mechanisms of IMI action involve the inhibition of the voltage sensitive ether-à-go-go potassium (EAG 1) channel activity in one of the mCRPC cell lines, DU145, reduction of cell proliferation, and induction of apoptosis [23].

Therefore, the aim of this study was to investigate the effects and mechanisms of IMI action during the migration and invasion in PC-3 cells, which can be considered as a cellular model of mCRPC because they are androgen-insensitive and have a high metastatic potential.

## 2. Results

### 2.1. IMI Inhibits Proliferation of PC-3 Cells without Causing Cell Death

The antiproliferative properties of IMI were evaluated by exposing the prostate cancer cell line PC-3 to different concentrations of IMI (3.8 × 10^−4^ to 1.0 × 10^2^ μM) for 12 h and 72 h. IMI treatment resulted in decreased proliferation of PC-3 cells in a dose-dependent and time-dependent manner (Figure 1a). In the IMI-treated group at 12 h, IMI at < 10 μM did not affect cell proliferation; however, concentrations above 10 μM resulted in decreased cell proliferation in a dose-dependent manner. In the IMI-treated group at 72 h, IMI had a dose-dependent inhibitory effect on cell proliferation over the entire dose range. To confirm cell viability, PC-3 cells were treated with the highest concentration (100 μM) of IMI for a longer period (72 h), and the cell viability was visualized using a LIVE/DEAD kit and quantified (Figure 1b,c). Although PC-3 cells were exposed to high doses for a long period, the number of live and dead cells and cell viability in IMI-treated cells did not show significant differences compared to those in the untreated cells. We concluded that IMI did not induce cell death but attenuated the proliferation of PC-3 cells.

### 2.2. IMI Inhibits PC-3 Cell Migration

To determine whether IMI inhibits migration in PC-3 cells, wound healing was monitored in PC-3 cells treated with different concentrations of IMI (6.25, 12.5, 25, 50, and 100 μM). As shown in Figure 2b,c, IMI significantly decreased the migration of PC-3 cells in a dose-dependent manner compared to that in the control, untreated cells (*p* < 0.001). These results demonstrated that IMI suppressed migration in PC-3 cells in vitro.

### 2.3. IMI Inhibits PC-3 Cell Invasion

To determine the effect of IMI on cell invasion, PC-3 cells were treated with 12.5, 25, 50, and 100 μM IMI, after which the cells were allowed to invade in Matrigel-coated Transwells for 24 h. The number of invading cells was significantly reduced by the IMI treatment in a dose-dependent manner (Figure 2c,d). Compared to the untreated group, IMI at 50 and 100 μM suppressed cell invasion by 79.8% and 92.5%, respectively. These data clearly show that IMI is a strong suppressor of PC-3 cell invasion.

### 2.4. IMI Inhibits Phosphorylation of Serine/threonine Protein Kinase (AKT), but Not that of Extracellular Signal-regulated kinase (ERK)1/2 in PC-3 Cells

In order to investigate the mechanism of IMI action, the effects of IMI treatment (6.25, 12.5, 25, 50, and 100 μM) on protein expression levels of AKT and ERK1/2 were evaluated using Western blot analysis in PC-3 cells. As shown in Figure 3, activation of AKT in PC-3 cells was significantly suppressed by IMI (6.25, 12.5, 25, 50, and 100 μM) at 48 and 72 h, as evidenced by the dose-dependent decrease in AKT phosphorylation. However, the expression of phosphorylated ERK1/2 was upregulated by IMI at 48 h and 72 h.

### 2.5. IMI Attenuates Nuclear Factor Kappa B (NF-κB) Signaling and the Expression of Proinflammatory Cytokines and Chemokines in PC-3 Cells

Cancer proliferation is associated with NF-κB signaling pathway and inflammatory cytokines. The protein expression levels of p-inhibitor of κB kinase (IKK)α/β, IKKα, p-inhibitor of κB (IκBα), IκBα, p-p65, p65, and β-actin in the NF-κB signaling pathway, which is downstream to AKT activation, and the secretion of inflammatory cytokines were estimated using Western blotting and RT-PCR, respectively. As a result, IMI inactivated the NF-κB signaling pathway in PC-3 cells (Figure 4a–e). Phosphorylation of IKKα/β, IκBα, and p65 was inhibited by IMI. Concurrently, degradation of IκBα was reduced by IMI. In particular, phosphorylated p65 levels were significantly attenuated in response to IMI treatment (25–100 μM) in a dose-dependent manner.

Because protein phosphorylation during NF-κB signaling regulates the expression of inflammatory cytokines, the messenger RNA (mRNA) expression of inflammatory cytokines, including tumor necrosis factor–*α* (*TNF-α*), interleukin-1β (*IL-1β*), and monocyte chemoattractant protein-1 (*MCP-1*) was quantitatively analyzed using qRT-PCR. The mRNA levels of all three cytokines were significantly suppressed by IMI treatment (Figure 4f–h). The mRNA level of *TNF-α* was significantly inhibited by 100 μM. The mRNA levels of *IL-1β* and *MCP-1* were significantly attenuated by IMI concentrations ranging from 12.5 μM to 100 μM and from 25 μM to 100 μM, respectively.

## 3. Discussion

This is the first study to demonstrate that IMI attenuates cell proliferation, migration, and invasion of PC-3 cells via inactivation of AKT-mediated NF-κB signaling and activation of ERK-mediated signaling. IMI suppressed the phosphorylation of AKT, IKKα/β, IκBα, and p65 in PC-3 cells in a concentration-dependent manner. Downregulated AKT-mediated NF-κB signaling was associated with decreased gene expression levels of cytokines including *TNF-α*, *IL-1β*, and *MCP-1*. Our findings suggest that imipramine could be a potential candidate for treatment of prostate cancers.

A number of signaling pathways regulate cellular metabolism and cancer progression. The representative mechanisms involved in cell survival, growth, proliferation, migration, and invasion in tumors and tumor cell lines include the phosphatidylinositol 3-kinase (PI3K)/AKT pathway and mitogen-activated protein kinase (MAPK)/extracellular signal-regulated kinase (ERK) pathway. These pathways are also important for mCRPC progression. PI3K activation stimulates downstream phosphatidylinositol (3,4,5)-trisphosphate (PIP3) production and further phosphorylation of AKT (Ser473 and Thr308) in the PI3K/AKT pathway [24]. PI3K inactivation by PI3K inhibitor (LY294002) attenuated the protein expression of PI3K (p85) and p-AKT (Ser473) and further suppressed PC-3 cell invasion [25]. Selective inhibition of the PI3K/AKT pathway with the AKT inhibitor, AZD5363, also retards cellular proliferation and tumor progression in CRPC xenografts [26]. In particular, AKT is an attractive therapeutic target to inhibit mCRPC progression because of the following characteristics of PC-3 cells: (1) excessive AKT level, (2) loss of phosphatase and tensin homolog (PTEN), and (3) AKT-induced inhibition of Raf/mitogen-activated protein kinase kinase (MEK)/ERK signaling cascades. According to phospho-proteomic analysis in metastatic tumor samples, mCRPC contains excessive amounts of AKT protein as the most common tyrosine kinase [27]. PTEN, a tumor suppressor gene that is negatively involved in AKT phosphorylation by PI3K, is lost in ~50% of tumor tissue in mCRPC patients and is known to inhibit cell proliferation and migration in prostate cancer [28,29]. Both excessive AKT accumulation and loss of PTEN accelerate prostate carcinogenesis. Raf/MEK/ERK signaling cascades are upregulated not only by Ras activation but also by AKT inhibition in several types of cancers, such as prostate and breast cancers [30,31]. ERK signaling has two opposing roles: one is associated with cell proliferation, migration, and invasion [32], and the other one is relevant to the proapoptotic action of chemotherapeutics. Various antitumor agents, including resveratrol, betulinic acid, apigenin, and ororidonin, trigger cancer cell death through ERK activation [33,34,35,36]. This is consistent with other chemotherapeutic agents in which phenethyl isothiocyanate activates MAPK/ERK signaling cascades and induces apoptosis in PC-3 cells [37]. In addition, upregulated MAPK/ERK signaling by AKT inactivation also induced loss of differentiation of PC-3 cells [38]. IMI inhibited AKT phosphorylation, increased ERK activation in PC-3 cells, and attenuated cell proliferation, instead of inducing cell death.

CRPC contains higher levels of AKT and NF-κB than CSPC and other cancers. The NF-κB family of proteins plays a pleiotropic role in controlling multiple cell functions such as proliferation, survival, cell death, invasion, and angiogenesis; however, this protein family plays a dichotomous role. The canonical NF-κB signaling pathway is executed through the activation of the IKK complex composed of the catalytic subunits (IKKα and IKKβ) and the regulatory subunit (IKKγ/NF-κB essential modifier, NEMO) [39]. Phosphorylated IKK activates IκBα proteins and then phosphorylates IκBα to release the p65(RelA)/p50(NF-κB1) complex, followed by degradation of IκBα by proteasome. The separated p65/p50 complex is translocated into the nucleus to activate gene transcription. Among NF-κB family proteins, not NF-κB/p50 but NF-κB/p65 is constitutively activated in human prostate adenocarcinoma. The NF-κB signaling pathway can be activated by AKT. Several studies have suggested that AKT/NF-κB signaling works together in prostate cancer cells [40,41]. The migration and invasion enhancer 1 (MIEN1) gene, which is highly expressed in prostate cancer cells, regulates AKT/NF-κB signaling. Phosphorylation of AKT stimulates mechanistic target of rapamycin complex 1 (mTORC1) and the IKK/NF-κB signaling cascade in PC-3 cells. Consistent with these cascades, inactivated AKT by IMI significantly attenuated the expression levels of phosphorylated IKKα/β, IκBα, and p65 in a dose-dependent manner in PC-3 cells. p65, binding to a consensus DNA sequence in the promoter region, regulates cancer metastasis [42]. Previous studies reported that blocking the p65 attenuated metastasis in prostate cancer; on the other hand, activation of p65 promoted invasion and migration of prostate cancer cells [43,44,45,46]. Moreover, suppressed AKT/NF-κB signaling by IMI affected downstream signaling cascades.

Proinflammatory cytokines and chemokines are regulated by NF-κB. Phosphorylation of p65 facilitates its binding to a specific DNA sequence, which triggers the transcriptional activation of NF-κB-regulated genes, including proinflammatory cytokine genes and chemokine genes. We found that IMI-induced deactivation of p65 also attenuated the expression levels of downstream genes, including two proinflammatory cytokine genes (*TNF-α* and *IL-1β*) and one chemokine gene (*MCP-1*). Prostate cancer cells express chemokine and chemokine receptors, as well as endogenously produced chemokines and cytokines such as *MCP-1*, *TNF-α*, *IL-1β*, IL-6, and IL-8 [47,48,49], some of which are secreted at a higher level in PC-3 cells (mCRPC cell line) than in LNCaP cells (mCSPC cell line). *MCP-1*, a member of the CC chemokine superfamily, is associated with recruitment and activation of monocytes during acute inflammation. Endogenous *MCP-1*, eliciting both autocrine and paracrine responses, promotes cell growth and invasion in prostate cancer cells including primary prostate epithelia (PrEC), LNCaP, C4-2B, and PC-3 cells [50]. *TNF-α* and *IL-1β*, the major proinflammatory cytokines associated with systemic inflammation, play a critical role in tumorigenesis, tumor progression, and carcinogenesis. Both cytokines also act as autocrine factors; thus, the secretion of endogenously produced *TNF-α* and *IL-1β* promotes cell proliferation and migration in PC-3 cells. In particular, it is proof that the expression of *IL-1β* is stimulated by p65 phosphorylation. Finally, IMI-induced suppression of the expression of *TNF-α*, *IL-1β*, and *MCP-1* may regulate autocrine signaling and inhibit cell proliferation, migration, and invasion.

In this study, we found that imipramine inhibits proliferation, migration, and invasion in a cellular model of mCRPC, which is expected to be related to AKT/NF-κB signaling. Some research suggested that upstream AKT signaling including PI3K, PDK1, and PTEN can be binding targets of imipramine [51], but further study will be needed to know the exact target of imipramine. In addition, it is necessary to observe the suppressive effect of imipramine on metastasis using a xenographic tumor model so that the results of in vitro studies can be confirmed in vivo.

## 4. Materials and Methods

### 4.1. Materials

IMI was purchased from Sigma-Aldrich (St. Louis, MO, USA). The antibody against β-actin was purchased from Santa Cruz Biotech (Santa Cruz, CA, USA). Antibodies against p-IKKα/β, IKKα, p-IκB, IκB, p-p65, p65, ERK1/2, p-ERK1/2 (Thr202/Tyr204), AKT, and p-AKT (Ser473) were purchased from Cell Signaling Technology (Beverly, MA, USA).

### 4.2. Cell Culture

The prostate cancer cell line PC-3 was obtained from the Korean Cell Line Bank (KCLB, Seoul, Korea) and cultured in Roswell Park Memorial Institute (RPMI) 1640 (Gibco, Grand Island, NY, USA) containing 10% fetal bovine serum (FBS; Gibco) and 100 U/mL penicillin/streptomycin (Gibco) at 37 °C and 5% CO_2_.

### 4.3. Cell Proliferation Assay

PC-3 cells were seeded in 96-well plates (1 × 10^3^ cells/well) and cultured overnight to allow cell adhesion. PC-3 cells were treated with 0.1% dimethyl sulfoxide (DMSO) or various concentrations of IMI (range: 3.8 × 10^−4^ to 1.0 × 10^2^ μM) for 12 h or 72 h. Subsequently, 10 μL of water-soluble tetrazolium salts, WST-1 reagent (Roche Diagnostics GmbH, Mannheim, Germany) was added to each well and incubated for 4 h at 37 °C in a humidified 5% CO_2_ atmosphere. Finally, the absorbance at 450 and 650 nm was measured using a SpectraMax Plus Plate Reader (Molecular Devices, Sunnyvale, CA, USA). PC-3 cells treated with 0.1% DMSO served as a control. The percentage of cell proliferation was calculated using the following equation:Cell proliferation = (OD_450_ − OD_650_ of IMI-treated cells)/(OD_450_ − OD_650_ of control) × 100.

### 4.4. Live/Dead Cell Viability Assay

Cell viability was confirmed using a LIVE/DEAD^®^ Cell Imaging Kit (Invitrogen, Waltham, MA, USA). PC-3 cells plated at a density of 1 × 10^3^ cells per well in a 96-well plate were treated with 0.1% DMSO or 100 μM IMI. After 72 h, calcein-AM (live, green) and propidium iodide (PI; dead, red) were added to the cells and incubated for 15 min at room temperature in the dark. Images of live and dead cells were acquired using a fluorescence microscope (Axio Observer A1; Carl Zeiss AG, Oberkochen, Germany). The percentage of cell viability was calculated according to the following formula:% of cell viability = (the number of viable cells/the total number of cells) × 100.

### 4.5. Wound Healing Assay

PC-3 cells were seeded at a density of 7 × 10^5^ cells/mL into the two wells of a culture insert (Ibidi GmbH, Martinsried, Germany). After the overnight incubation, cells were grown to full confluence, and the culture inserts were removed to make the wound gap. Cells were washed with phosphate buffered saline (PBS) and incubated with five different concentrations of IMI (0–100 μM) for 12 h. Representative images for each concentration were captured at 10× magnification using a digital camera attached to an inverted microscope to quantify the relative migration of cells (Axio Observer A1). The area of the wound was quantitatively analyzed using MetaMorph software (Molecular Devices, Sunnyvale, CA, USA), and the percentage of wound recovery was evaluated. Coverage in untreated PC-3 cells was defined as 100%.

### 4.6. Cell Invasion Assay

The cell invasion assay was performed using 24-well Transwell chambers with polycarbonate filters with 8 μm pore size (Corning Costar, Corning, New York, NY, USA). Transwell chambers were coated with gelatin solution (0.1%) on the lower surface and Matrigel on the upper surface for 30 min. PC-3 cells were seeded at a density of 2 × 10^4^ cells/mL in serum-free medium onto the upper compartment of the Transwell, and the lower chambers were filled with medium containing 10% FBS and various concentrations of IMI (12.5, 25, 50, and 100 μM). After 24 h, the noninvaded cells and the Matrigel on the upper chamber were gently removed using a cotton-tipped swab; then, cells at the undersurface of the filters were stained with crystal violet stain solution for 20 min and rinsed several times in distilled water. The number of invaded cells was quantified by visual counting after being photographed using an inverted microscope at 10× magnification.

### 4.7. Western Blot Analysis

PC-3 cells (4 × 10^5^ cells/dish) were seeded in 100 mm dishes for 24 h and starved with 0.05% FBS overnight. PC-3 cells were incubated with five different concentrations of IMI (3.25, 12.5, 25, 50, and 100 μM) for 48 h or 72 h. After the treatment, the cells were lysed using cell lysis buffer (Cell Signaling Technology, Beverly, MA, USA) containing a protease inhibitor cocktail from Roche and phosphatase inhibitor (Sigma Aldrich) for 30 min on ice. After centrifugation, a bicinchoninic acid (BCA) protein assay kit (Thermo Scientific, Rockford, IL, USA) was used to determine the protein concentration. Equal amounts of protein lysate were separated in 10% sodium dodecyl sulfate (SDS) polyacrylamide gels and transferred to a polyvinylidene difluoride (PVDF) membrane (Bio-Rad Inc., Hercules, CA, USA). The membrane was blocked with 5% skim milk in Tris-buffered saline with 0.1% Tween-20 (TBST) for 1 h at room temperature and then incubated with the specific primary antibodies, including AKT, phospho-AKT (Ser473), ERK1/2, phospho-ERK1/2 (Thr202/Tyr204), phospho-IKKα/β, IKKα, phospho-IκB, IκB, phospho-p65, and p65, overnight at 4 °C. The membranes were washed in TBST three times and incubated with horseradish peroxidase (HRP)-conjugated secondary antibody at room temperature for 2 h. Protein bands were visualized using a chemiluminescence detection kit (ATTO, Tokyo, Japan). Beta-actin was used as the loading control.

### 4.8. Total RNA Isolation and Quantitative Reverse-Transcription Polymerase Chain Reaction (RT-qPCR)

Total RNA was isolated using the NucleoPin^®^ RNA Plus XS Kit (Macherey-Nagel, Bethlehem, PA, USA) according to the manufacturer’s instructions. Reverse transcription of RNA was performed using the ReverTra Ace^®^ qPCR RT Master Mix (Toyobo, Osaka, Japan). First-strand complementary DNA (cDNA) was prepared using 1 μg of total RNA. The real-time PCR reaction was performed in a volume of 20 μL containing 0.1 μg of cDNA, 1 μM of each primer (Table 1), and Power SYBR^®^ Green PCR Master Mix (Applied Biosystems, Carlsbad, CA, USA). Thermal cycling was carried out in a StepOnePlus Real-Time PCR system (Applied Biosystems, Carlsbad, CA, USA) with a program of 95 °C for 5 min, followed by 40 cycles of denaturation at 95 °C for 5 s annealing, and elongation at 60 °C for 10 s. Gene expression levels were normalized to the expression level of the housekeeping gene glyceraldehyde-3-phosphate dehydrogenase (*GAPDH*). Relative gene expression changes, calculated using the 2^−∆∆CT^ method, were reported as number-fold changes compared to those in the control samples.

### 4.9. Statistics

All experiments were independently performed at least three times and the values were expressed as the mean ± standard error of the mean (SEM). Statistical analysis was performed using GraphPad Prism version 5 (GraphPad Software, Inc., La Jolla, CA, USA). Statistical significance was evaluated by unpaired *t*-test for comparison between two groups and one-way ANOVA followed by Dunnett’s post hoc test for multiple groups. A *p*-value < 0.05 was considered to indicate statistically significant difference.

## 5. Conclusions

In conclusion, our results demonstrated that IMI treatment inhibited cell proliferation, migration, and invasion in mCRPC PC-3 cells. The suggested mechanisms of IMI in PC-3 cells include the modulation of AKT/ERK signaling and suppression of the AKT/NF-κB signaling pathway by preventing IκBα degradation, blocking p65 phosphorylation, and regulating chemokine and cytokine production. We suggest that IMI may be a potential chemotherapeutic candidate against metastatic CRPC.

## Figures and Tables

**Figure 1 molecules-25-04619-f001:**
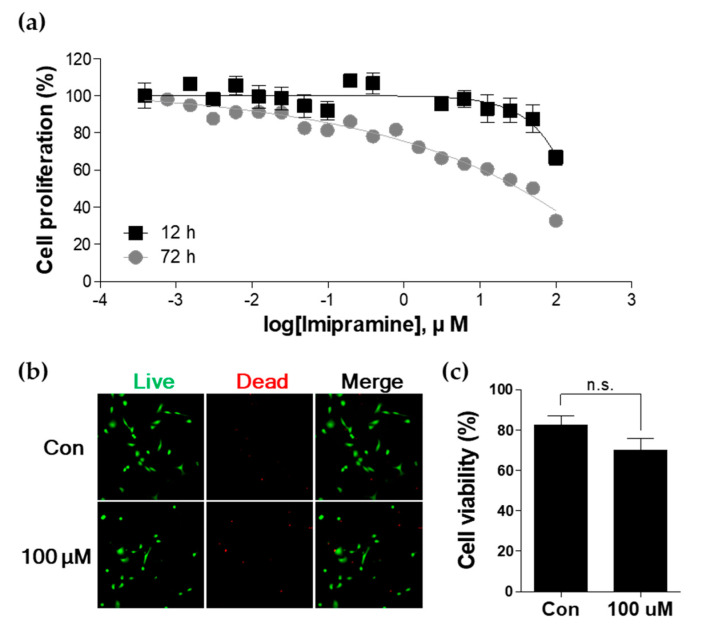
Effect of imipramine treatment on proliferation of PC-3 cells. (**a**) PC-3 cells were treated with 3.8 × 10^−4^ to 1.0 × 10^2^ μM imipramine (IMI) for 12 h and 72 h, and cell proliferation was evaluated using a water-soluble tetrazolium salts, WST-1 assay. (**b**,**c**) Cell viability at 100 μM of imipramine treatment for 72 h was visualized and quantified using a LIVE/DEAD kit. Green and red fluorescence indicates live and dead cells, respectively. n.s.: not significant.

**Figure 2 molecules-25-04619-f002:**
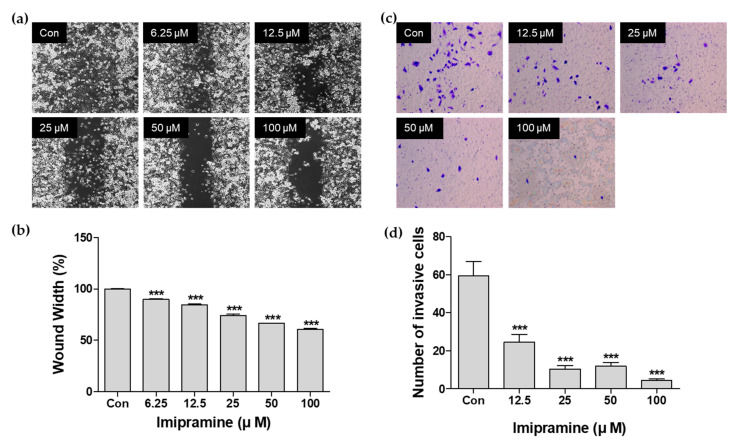
Inhibition of cell migration and invasion by imipramine. (**a**) PC-3 cells were treated with imipramine (0–100 μM) for 12 h. Representative images of the cells treated with the indicated doses were acquired in an optical microscope after the wounding. (**b**) The percentage of wound recovery was normalized to untreated control cells. (**c**) PC-3 cells were treated with imipramine (0–100 μM) for 12 h and the effects of imipramine on invasion were analyzed using Matrigel invasion assays. Representative images were taken in the bottom of Transwell filter at indicated time points via phase-contrast microscopy. (**d**) Quantitative data of an invasion assay were expressed as the number of invasive cells. Data are presented as the mean ± standard error of the mean (SEM) and analyzed using one-way ANOVA; *** *p* < 0.001 compared with untreated cells.

**Figure 3 molecules-25-04619-f003:**
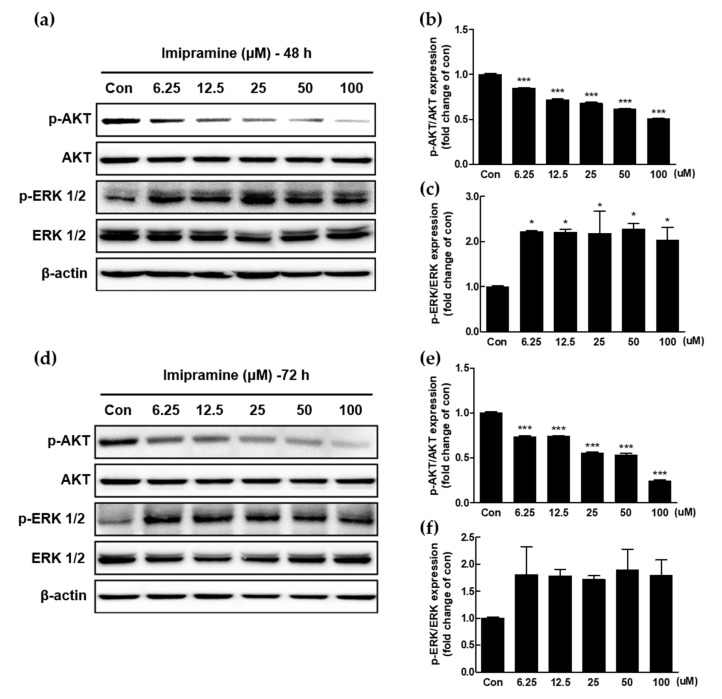
Effect of imipramine on phosphorylation of serine/threonine protein kinase (AKT) and extracellular signal-regulated kinase (ERK)1/2 at 48 h and 72 h in PC-3 cells. (**a**,**d**) Protein expression of AKT, p-AKT, ERK1/2, p-ERK1/2, and β-actin in PC-3 cells was detected using Western blotting. (**b**,**c**,**e**,**f**) Quantitative data of phosphorylated AKT and ERK were normalized to the untreated cells. Data are presented as the mean ± SEM (*n* = 3) and analyzed using one-way ANOVA; * *p* < 0.05, and *** *p* < 0.001 compared with untreated cells.

**Figure 4 molecules-25-04619-f004:**
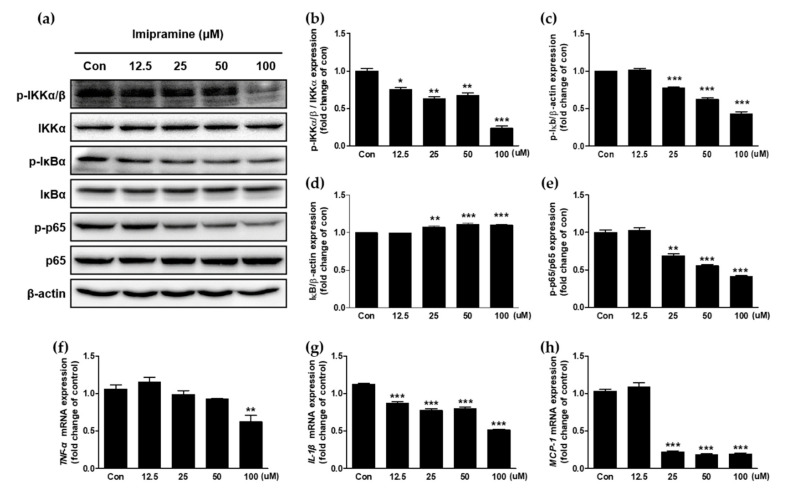
Effect of imipramine on protein phosphorylation during nuclear factor kappa B (NF-κB) signaling and the expression of inflammatory cytokines at 48 h in PC-3 cells. (**a**) Protein expression of p-inhibitor of κB kinase (IKK)α/β, IKKα, p-inhibitor of κB (IκBα), IκBα, p-p65, p65, and β-actin in PC-3 cells was detected using Western blotting. (**b**) Quantitative data of phosphorylated IKKα/β were normalized to the untreated cells. (**c**) Quantitative data of phosphorylated IκBα were normalized to the untreated cells. (**d**) Quantitative data of IκBα were normalized to the untreated cells. (**e**) Quantitative data of phosphorylated p65 were normalized to the untreated cells. (**f**–**h**) The production of tumor necrosis factor–*α* (*TNF-α*), interleukin-1β (*IL-1β*), and monocyte chemoattractant protein-1 (*MCP-1*) messenger RNA (mRNA) was analyzed using quantitative real-time RT-PCR. Data are presented as the mean ± SEM of three independent experiments and analyzed using one-way ANOVA; * *p* < 0.05, ** *p* < 0.01, and *** *p* < 0.001 compared with untreated cells.

**Table 1 molecules-25-04619-t001:** Primer sequences for RT-qPCR.

Species	Gene	Primer Sequence	
		Forward	Reverse
**Human**	*TNF-α*	CCTCTCTCTAATCAGCCCTCTG	GAGGACCTGGGAGTAGATGAG
	*IL-1β*	ATGATGGCTTATTACAGTGGCAA	GTCGGAGATTCGTAGCTGGA
	*MCP-1*	CAGCCAGATGCAATCAATGCC	TGGAATCCTGAACCCACTTCT
	*GAPDH*	GGAGCGAGATCCCTCCAAAAT	GGCTGTTGTCATACTTCTCATGG

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
