# Peer review of "Imipramine Inhibits Migration and Invasion in Metastatic Castration-Resistant Prostate Cancer PC-3 Cells via AKT-Mediated NF-κB Signaling Pathway"

_molecules, 2020, doi:10.3390/molecules25204619_

Round 1
Reviewer 1 Report
This manuscript describes initial cell culture studies on the effects of imipramine on the PC-3 cell line. The study is not extensive, however the results are interesting and for that reason worthy of publication. There are a number of minor English language errors in the manuscript that I have listed below, as well as a few methodological questions. My major reservation with the manuscript was that the Discussion section did not seem to address the manuscript and I feel that it could be re-written to better reflect the findings of the study.
- Line 43: “by decreasing” should be “as evidenced by decreased circulating levels of”
- Line 55: delete “the”
- Line 64: “suppressed the activation of Shh-IL6-RANKL signalling”
- Line 65: “metastasis of CRPC cells”
- Line 66: “derived by chronic…”
- Line 69-70: “mechanisms of action of TCAs in mCRPC”
- Line 74: delete “cells (end of line), add the word “cells” after prostate cancer in line 75
- Line 76-77: “there was a positive correlation between adjusted odds ratio and IMI” There is something missing from this sentence fragment. Odds ratio for what? (and IMI use)?
- Lines 83-84: “which can be considered as a mCRPC cell line because they are androgen-insensitive and have a high metastatic potential.”
- Line 87: “the prostate cancer cell line”
- Line 90: “At 12 h, IMI at <10µM did not…”
- Line 91-92: “At 72 h, IMI had a…”
- Line 92: “on cell proliferation” (delete “the”)
- Line 99: “on proliferation of PC-3 cells”
- The Live/Dead kit will detect adherent cells, whereas dead cells may have detached from the culture surface and lysed. Please confirm that IMI-treated cultures did not accumulate increasing (light microscopic) evidence of detached cells.
- Line 107: “decreased migration of PC-3 cells”
- Line 109: “in vitro” should be in italics.
- Cell migration is not a substitute for metastasis, it is just cell migration. (Sometimes a combination of in vitro cell migration and invasion assays are used to suggest that a cell line may have metastatic potential in vivo, however, this is usually presented in correlation with in vivo assays). It is suggested that this terminology is removed and that wording similar to the next section (2.3) is used.
- Line 114: “assays”
- Line 117: “analysed using one-way…”
- It is noticeable that imipramine concentrations used in this study ranged from 6.25/12.5 - 100µM, which is up to 100 times that achievable as a therapeutic agent in vivo. How translatable do the investigators feel their study results are in light of this discrepancy? Will imipramine have any effects at (in vivo) pharmacologically achievable concentrations?
- As PC-3 cells are human in origin (not mouse), it is more appropriate to write AKT (not Akt).
- Lines 130-131: The wording here is difficult to follow. Suggested alternative: “…. as evidenced by the dose-dependent decrease in AKT phosphorylation.” (not “which was accompanied with…”)
- Figures 3 and 4: Please check all western blots. It appears that some of the blots for total and phosphorylated ERK/AKT do not match. (Immunodetection of pAKT and total AKT should be performed on the same blot. Similarly, detection of pERK and total ERK should be performed on the same blot, etc).
- The y-axis label in Figure 4b appears to have an error (should be p-IKKα/β / IKKα (not IKKβ) according to the western blot label and Methods section).
- Line 168: delete “protein expression of” (redundant)
- Note that when referring to proteins, use normal lettering. When referring to mRNA or DNA, use italics. For example, TNF-α refers to the protein, TNF-α refers to the mRNA or DNA (gene). This convention should be applied throughout the document (text and figures).
- Lines 169-170: Based on the experiments performed by the authors, it is not possible to conclude that downregulation of NF-κB signalling was the cause of the decrease in cytokine expression. A suggested alternative is to state that it was associated with this decrease.
- Lines 170-171: I do not feel that the downregulation of MCP-1 was dose-dependent.
- Lines 171-172: The sentence “The following mechanisms are in agreement…” does not make sense in this context and can be deleted.
- I do not understand why the authors have focussed their discussion on AKT/MAPK/NF-κB signalling in prostate cancer, topics that are very well studied and reviewed elsewhere, rather than imipramine and its potential activity as an anti-cancer agent. Although the Discussion section reads well, it does not seem to be related to the findings of the study, the significance of results, or on potential future directions. If possible, I would suggest re-writing the Discussion to better highlight the importance of this study. In addition, the authors have not addressed potential mechanisms for imipramine action related to drug binding. With use of such high concentrations of imipramine, is it possible that the effects that they are seeing are non-specific?
- Note that PC-3 is the original and ATCC spelling of this cell line (not PC3).
- Lines 248-253: There is an error here that requires amendment. It appears that the authors have not deleted manuscript template instructions, which are inserted in the middle of their own method.
- Line 288: “cotton-tipped” (not cotton-tripped)
- Line 298: ”Suggested alternative: “Equal amounts of protein lysate were separated in 10% sodium dodecyl sulfate (SDS) polyacrylamide gels and transferred to…” (The abbreviation SDS-PAGE is not used elsewhere in the document).
- It seems as though the authors are only performing quantitative PCR, not quantitative reverse transcription. For this reason, RT-qPCR is the more accurate terminology.
- Line 333: The investigators have not investigated p65 translocation. As such, the statement that imipramine blocked p65 translocation is not warranted and should be deleted.
- Formatting of references should be checked for consistency, in particular journal names.
Reviewer 2 Report
Taking into account the relationship found between treatment with tricyclic antidepressants and a lower progression of prostate cancer, the authors set out to study the effects and mechanisms of treatment with Imipramine (IMI) in a cellular model of castration-resistant prostate cancer. The results show that IMI has an antiproliferative effect on PC3 cells, without causing their death. Through Matrigel invasion and wound healing assays, the authors demonstrate that IMI is capable of affecting the migratory and invasive capacity of PC3 cells in vitro. On the other hand, IMI decreases the expression of the phosphorylated form of Akt, which suggests the inhibition of their signaling cascade. Since the NF-kappaB signaling cascade is found downstream of Akt, the association of which to tumor proliferation is well known, the authors studied the levels of phosphorylation of IKK, IkappaB and p65. They found that with IMI treatment the expression of the phosphorylated form of the kinase that activates the cascade (IKK), the phosphorylated form of the inhibitor of NF-kappaB (IkappaB) and the phosphorylated form of NF-kappaB (p65) decreased. Taken together, this strongly suggests this signaling pathway is inhibited due to treatment, which is corroborated by observing the decrease in the expression of target genes such as the inflammatory cytokines TNF alpha, IL1beta and MCP-1.
The authors conclude that IMI treatment inhibits cell proliferation, migratory and invasive capacity of PC3 cells through a mechanism that includes inhibition of the Akt / NF-kappaB signalling pathway.
The writing of this paper makes reading easy and fluent. It's easy to follow the stream of thoughts and ideas that lead from one experiment to the next. The work is well organized and the results are presented clearly and concisely.
It is an original work that provides new evidence and that can contribute to the development of new antitumor therapies.
My recommendation is in favor of its publication. However, I do have some minor criticisms that I hope are constructive:
1) In line 84 they mention PC3 as an androgen insensitive cell line. It would be preferable to say that they are a cellular model of mCRPC and voila.
2) On line 109 it is state that IMI suppresses metastasis in PC3 cells. To support this claim, it would be necessary to have in vivo experiments in which xenographic tumors are generated and the metastatic capacity of the cells is evaluated. In my opinion, I would say that the treatment affects the migratory and invasive capacity of cells in vitro, which is an incentive to think that it can affect metastasis in vivo.
